# Shapley Value-based Contrastive Alignment for Multimodal Information Extraction

## ABSTRACT

The rise of social media and the exponential growth of multimodal communication necessitates advanced techniques for Multimodal Information Extraction (MIE). However, existing methodologies primarily rely on direct Image-Text interactions, a paradigm that often face the significant challenges due to semantic and modality gaps between images and text. In this paper, we introduce a new paradigm of Image-Context-Text interaction, where large multimodal models (LMMs) are utilized to generate descriptive textual context to bridge these gaps. In line with this paradigm, we propose a novel Shapley Value-based Contrastive Alignment (Shap-CA) method, which aligns both context-text and context-image pairs. Shap-CA initially applies the Shapley value concept from cooperative game theory to assess the individual contribution of each element in the set of contexts, texts and images towards total semantic and modality overlaps. Following this quantitative evaluation, a contrastive learning strategy is employed to enhance the interactive contribution within context-text/image pairs, while minimizing the influence across these pairs. Furthermore, we design an adaptive fusion module for selective cross-modal fusion. Extensive experiments across four MIE datasets demonstrate that our method significantly outperforms existing state-of-the-art methods. Code will be released upon acceptance.

## CCS CONCEPTS

• **Computing methodologies → Information extraction**.

## KEYWORDS

multimodal information extraction, multimodal alignment, contrastive learning

## 1 INTRODUCTION

The exponential growth of social media platforms has initiated a new phase of communication, characterized by the exchange of multimodal data, primarily texts and images. This diverse landscape necessitates advanced techniques for multimodal information extraction (MIE) [10, 39, 45], which primarily aims to utilize auxiliary image inputs to enhance the performance of identifying entities or relations within the unstructured text.

*ACM MM, 2024, Melbourne, Australia*

© 2024 Copyright held by the owner/author(s). Publication rights licensed to ACM.
ACM ISBN 978-x-xxxx-xxxx-x/YY/MM
https://doi.org/10.1145/nnnnnnn.nnnnnnn

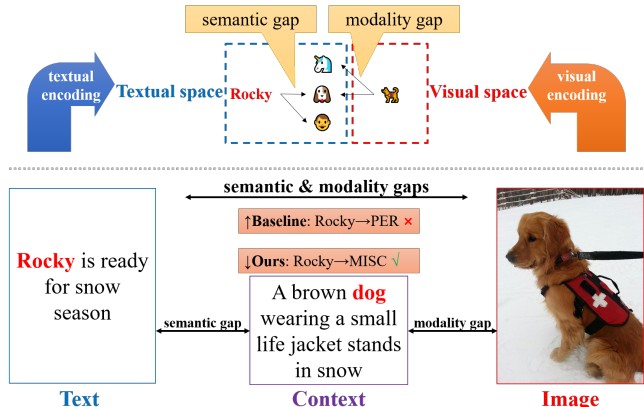

**Figure 1: The semantic and modality gaps.**

To the best of our knowledge, the majority of previous methods of MIE mainly concentrate on the direct interaction between images and text. These approaches either (1) encode images directly and employ efficient attention mechanisms to facilitate image-text interactions [4, 30, 34, 37, 41, 43, 44], or (2) detect finer-grained visual objects within images and use Graph Neural Networks or attention mechanisms to establish interactions between text and objects [9–11, 17, 18, 42, 45, 46, 51, 53]. Despite the advancements made by these methods, this direct Image-Text interaction paradigm still suffers from the simultaneous existence of semantic and modality gaps. The semantic gap refers to the disparity between the meaning conveyed by the text and the actual content depicted by the image. For instance, in Figure 1, the word "Rocky" could refer to the dog in the image but could also be a name for a cat or a person. Furthermore, even the direct alignment of the word "dog" with an image of a dog remains a challenge due to the representation inconsistency, resulting in a modality gap. The presence of these two gaps weakens the connection between images and text, potentially leading to erroneous predictions of entities or relations.

To mitigate the semantic and modality gaps, we propose an approach that relies on an intermediary, rather than direct interactions between texts and images. Given that text is the primary source of information extraction and images serve a supporting role, we suggest the use of a textual intermediary to bridge these gaps. As large multimodal models (LMMs) demonstrate impressive results on instruction-following and visual comprehension [26, 27], and excel at generating descriptive textual context for images, we employ the powerful generative ability of LMMs to create an intermediate context for reducing the burden of aligning image and text directly. As shown in Figure 1, with the divide-and-conquer strategy, we only need to model the connections between the generated context and the original text or image, forming an Image-Context-Text interaction paradigm.

The core challenge now is aligning the context-text pairs and context-image pairs in the hidden feature space, respectively, where the alignment with text bridges the semantic gap, and the alignment with images addresses the modality gap. In this scenario, contexts, texts, and images can be considered as three kinds of elements in the hidden space trying to achieve the optimal collective semantic and modality overlaps. Drawing inspiration from cooperative game theory, we adapt the concept of the Shapley value [14], which provides a equitable mechanism for assessing each individual's contribution towards collective utility. In this paper, we propose a novel Shapley Value-based Contrastive Alignment (**Shap-CA**) method, which leverage a contrastive learning strategy to construct coherent and compatible representations for these three elements based on their Shapley values. Within this framework, each element is considered as a player, and the collective utility of a set of elements is defined as the total semantic or modality overlap. Shap-CA initially calculates the average marginal contribution of each player to the collective utility to estimate their Shapley value. Intuitively, contexts contributing significantly to the overall overlap (i.e., having a larger Shapley value) should have a higher probability of forming a true pair with the text or image. Consequently, a contrastive learning strategy is then employed to enhance the interactive contribution within context-text/image pairs, while minimizing the influence across these pairs. Through this process, Shap-CA not only strengthens the intrinsic connections within each pair but also accentuates the disparities between different pairs, effectively bridging the semantic and modality gaps. Furthermore, we design an adaptive fusion module to obtain the informative fused features across modalities. This module assesses the relevance of each modal feature to the bridging context, strategically weighting their importance to achieve a finer-grained selective cross-modal fusion. Finally, a linear-chain CRF [22] or a word-pair contrastive layer is employed for prediction.

Overall, the main contributions of this paper can be summarized as follows:

- We are the first to introduce the Image-Context-Text interaction paradigm and leverage LMMs to generate descriptive context as a bridge to mitigate semantic and modality gaps for MIE.
- We propose a novel Shapley value-based contrastive alignment method, capturing semantic and modality relationships within and across image-text pairs for effective multimodal representations.
- Extensive experiments demonstrate that our method substantially outperforms existing state-of-the-art methods on four MIE datasets.

## 2 RELATED WORK

**Multimodal Information Extraction** Multimodal Information Extraction (MIE) is an evolving research domain primarily aimed at enhancing the recognition of entities and relations by utilizing supplemental image inputs. This field can be primarily subdivided into three critical tasks: Multimodal Named Entity Recognition (MNER), Multimodal Relation Extraction (MRE) and Multimodal Joint Entity-Relation Extraction (MJERE). In particular, the tasks of **MNER** [37, 39] and **MRE** [10, 51] are concerned with identifying

entities and relations separately, while the **MJERE** task[45] aims to extract entities and their associated relations jointly. Most of existing methods are concentrated around the paradigm of direct Image-Text interaction[9, 30, 34, 37, 42–45, 51, 53]. Wang et al. [41] designed a fine-grained cross-modal attention module to enhance the cross-modal alignment. Sun et al. [37], Xu et al. [43], and Liu et al. [28] focus on quantifying and controlling the influence of images on texts through gate mechanisms or text-image relationship inference. To facilitate the alignment between visual objects and text, Zhao et al. [50] and Yuan et al. [45] propose a heterogeneous graph network and an edge-enhanced graph neural network, respectively. Additionally, there are some other studies [18, 39] that employ external knowledge, such as machine reading comprehension [18], to foster model reasoning. Regardless of these substantial advancements, these approaches still fail to address the potential dual gaps existing between images and texts due to the representation inconsistencies. In this study, we aim to comprehensively consider these semantic and modality gaps and propose Shap-CA, which leverages the bridging context to mitigate these gaps and achieve a more coherent and compatible representation.

**Large Multimodal Models** Large multimodal models have recently gained substantial traction in the research community [2, 23]. Similar to the trend observed with large language models, several studies have indicated that scaling up the training data [6, 12, 49] or model size [6, 31] can significantly enhance these large multimodal models' capabilities. Moreover, visual instruction tuning can equip large multimodal models with excellent instruction-following, visual understanding, and natural language generation abilities [27]. This advancement empowers these models to excel in interpreting images according to instructions and generating informative textual contexts. However, their open-ended generative characteristics have resulted in less than satisfactory performance when directly applied to information extraction tasks [38].

## 3 METHODOLOGY

### 3.1 Task Definition

Given an input text $t = \{t_1, \ldots, t_{n_t}\}$ with $n_t$ tokens and its attached image $I$, our method aims to predict the output entity/relation labels $y$. The format of the labels $y$ is task-dependent. Specifically, for MNER, they are sequential labels. For MRE and MJERE, they are word-pair labels [45].

### 3.2 Overview

The architecture of Shap-CA is shown in Figure 2. Initially, we utilize a LMM, e.g., LLaVA-1.5 [26], to extract the textual bridging context from the image. Subsequently, we employ a pretrained textual transformer to extract features from the text and the context. In parallel, an image encoder is used to derive visual features from the image. Following this, we apply a Shapley value-based contrastive alignment to construct more coherent representations. The adaptive fusion module is then employed to obtain the informative features across modalities. Finally, these comprehensive representations are fed into a CRF or a word-pair contrastive layer for prediction.

**Figure 2: The overall architecture of Shap-CA.**

## 3.3 Encoding Module

**Context Generation** Given an image $I$, we utilize a pretrained LMM to generate a textual context as the bridging context $c = \{c_1, \ldots, c_{n_c}\}$ with $n_c$ tokens.

**Textual and Visual Encoding** In order to acquire contextualized textual features, we concatenate the input text $t$ with the bridging context $c$. Following this, we employ a pretrained textual transformer to extract both sentence-level and token-level features:

$$x_t, H_t, x_c, H_c = \text{Transformer}([t; c]) \quad (1)$$

where $x_t, x_c \in \mathbb{R}^d$ represent the sentence-level features, while $H_t \in \mathbb{R}^{n_t \times d}, H_c \in \mathbb{R}^{n_c \times d}$ denote the token-level features of the input text and context, respectively. Simultaneously, we employ an image encoder to extract the visual features of the image $I$:

$$x_v, H_v = \text{ImageEncoder}(I) \quad (2)$$

where $x_v, H_v$ denotes the global visual feature and the regional visual features of the image, respectively.

## 3.4 Shapley Value-based Contrastive Alignment

To construct more coherent representations, we leverage the Shapley value to perform both context-text alignment and context-image alignment. The Shapley value offers a solution for the equitable allocation of total utility among players based on their individual marginal contributions, and has widespread application [15, 16, 20]. We are the first to utilize the Shapley value for multimodal alignment through contrastive learning.

### 3.4.1 Preliminary.
In the context of a cooperative game, suppose we have $k$ players, represented by $K = \{1, \ldots, k\}$, and a utility function $u : 2^k \rightarrow \mathbb{R}$ that assigns a reward to each coalition (subset) of players. The Shapley value of player $i$ is then defined as [19]:

$$\phi_i(u) = \frac{1}{k} \sum_{S \subseteq K \backslash \{i\}} \frac{1}{\binom{k-1}{|S|}} [u(S \cup \{i\}) - u(S)] \quad (3)$$

The Shapley value essentially quantifies the average marginal contribution of a player to all potential coalitions (subsets). We detail the context-text alignment as an example as follows.

### 3.4.2 Context-Text Alignment.

*Shapley Value Approximation.* In the context-text alignment, inputs are a mini-batch of $k$ context-text pairs $\{(x_c{}^a, x_t{}^a)\}_{a=1}^k$. Here, we view the $k$ bridging contexts as players, denoted as $K = \{1, \ldots, k\}$ for simplicity. These players, or contexts, collaboratively contribute to the semantic comprehension of a specific text feature. Consider the $j$-th pooled text feature, $x_t{}^j$, and a selected subset of context players, denoted as a coalition $S \subseteq K$. The central idea is based on an assumption: if all the contexts within the subset $S$ and the text $x_t{}^j$ form positive pairs, the utility of $S$ for $x_t{}^j$ would be represented by the expected semantic overlap between them. This utility captures the collective semantic relationships between the text and the contexts within the coalition, as formalized by:

$$u_j(S) = \sum_{i \in S} p_i \text{sim}(x_t{}^j, x_c{}^i)$$

$$p_i = \frac{e^{\text{sim}(x_t{}^j, x_c{}^i)/\tau}}{\sum_{a \in S} e^{\text{sim}(x_t{}^j, x_c{}^a)/\tau}} \quad (4)$$

Here, $\text{sim}(x_t{}^j, x_c{}^i)$ denotes the semantic overlap between the text and each context (i.e., individual semantic contribution), measured by cosine similarity. The weight $p_i$, computed through a softmax operation with a temperature of $\tau$, models the cooperative behavior among different contexts by normalizing these individual semantic contribution. This approach intuitively suggests that the stronger the semantic overlap a context shares with the text (i.e., the larger

the semantic contribution), the more likely it is to form a true pair with the text in real-world situations. From this perspective, the utility of the coalition can be interpreted as an expectation over the semantic overlaps of each context within $S$ with the text, where the expectation weights are given by the likelihood of each context-text pair being positive. This method naturally prioritizes contexts that have a higher degree of semantic overlap with the text, thereby refining the overall semantic understanding.

However, as indicated by Eqn. 3, the computation of the Shapley value requires an exponentially large number of computations relative to the size of the mini-batch, which poses a challenge during training. To address this, we extend Monte-Carlo approximation methods [8, 33] to our training setting for estimating the Shapley value. We present the detailed algorithm in Alg. 1. It begins with a random permutation of $k$ context players and an initial stride of $s$, which is set to batch_size/2. At each iteration, the algorithm scans each player in the current permutation and calculates the marginal contribution when the player is added to the coalition formed by the preceding players. This marginal contribution serves as an unbiased estimate of the Shapley value for that player, improving with each iteration. Subsequently, the algorithm updates the permutation through a cyclic shift of the current stride and halve the stride for the next iteration. This gradual reduction in stride results in more stable marginal contributions over time, as the size of the subsets for each player changes less dramatically on average. The process continues until the stride falls to zero.

---

**Algorithm 1:** Cyclic Shapley Value Approximation

---

**INPUT**: $k$ context players, a pooled text $x_t{}^j$, an initial stride $s$ = batch_size/2

**OUTPUT**: Approximated Shapley value of $k$ players $\{\hat{\phi}_1(u_j), \ldots, \hat{\phi}_k(u_j)\}$

$P \leftarrow$ Random permutation of the players;

**while** $s > 0$ **do**
  **for** $i \in \{1, \ldots, k\}$ **do**
    Compute the marginal contribution $m_{P[i]}$ when adding player $P[i]$ to the preceding coalition $P[1 \sim i-1]$;
    Update $\hat{\phi}_{P[i]}(u_j)$ with $m_{P[i]}$;
  **end**
  Apply a cyclic shift to $P$ by $s$ steps;
  $s = s/2$ ;
**end**

---

*Contrastive Learning.* After acquiring the approximated Shapley value $\{\hat{\phi}_1(u_j), \ldots, \hat{\phi}_k(u_j)\}$, we introduce a context-to-text contrastive loss that aims to maximize the average marginal semantic contribution from each context player to the text with which it forms a true positive pair (i.e., the text from the same pair), while simultaneously minimizing the contributions between the true negative pairs (i.e., all other texts not paired with this context).

$$\mathcal{L}_{c2t} = -\frac{1}{k} \sum_{j=1}^{k} \left[ \hat{\phi}_j(u_j) - \sum_{i \neq j} \hat{\phi}_i(u_j) \right] \quad (5)$$

In a symmetrical manner, we can treat the $k$ pooled text as players, and derive the text-to-context contrastive loss $\mathcal{L}_{t2c}$. The semantic loss $\mathcal{L}_{semantic}$ is the average of these two contrastive losses:

$$\mathcal{L}_{semantic} = \frac{1}{2}(\mathcal{L}_{c2t} + \mathcal{L}_{t2c}) \quad (6)$$

*3.4.3 Context-Image Alignment.* Similarly, we can employ Alg. 1 to estimate the Shapley value for a mini-batch of $k$ context-image pairs $\{(x_c{}^a, x_v{}^a)\}_{a=1}^{k}$ and derive the context-to-image loss $\mathcal{L}_{c2v}$ and the image-to-context loss $\mathcal{L}_{v2c}$. The modality loss $\mathcal{L}_{modality}$ is the average of these two contrastive loss:

$$\mathcal{L}_{modality} = \frac{1}{2}(\mathcal{L}_{c2v} + \mathcal{L}_{v2c}) \quad (7)$$

## 3.5 Adaptive Fusion

To facilitate a finer-grained fusion across modalities, we develop an adaptive attention fusion module. This module dynamically weights the importance of different features in two modalities, based on their relevance to the context that connects them. Given the representations $H_t, H_c, H_v$ obtained from Section 3.3, we initially employ linear projection to transform them into a set of matrices: a query matrix $Q_v \in \mathbb{R}^{n_v \times D}$, a key matrix $K_t \in \mathbb{R}^{n_t \times D}$, a value matrix $V_t \in \mathbb{R}^{n_t \times D}$, and a context matrix $C \in \mathbb{R}^{n_c \times D}$. Subsequently, we calculate the bridging term $B$ as follows:

$$B = \text{mean}\left(\frac{C^\top C}{\sqrt{D}}\right) \in \mathbb{R}^D \quad (8)$$

This bridging term is then employed to dynamically modify the content of queries $Q_v$ and keys $K_t$ based on their relevance to the bridging term:

$$Q_v' = g_q \odot Q_v + (1 - g_q) \odot B$$
$$K_t' = g_k \odot K_t + (1 - g_k) \odot B \quad (9)$$

where $\odot$ represents the Hadamard product, $g_q \in \mathbb{R}^{n_v \times D}$ and $g_k \in \mathbb{R}^{n_t \times D}$ are gating vectors that are used to capture the relevance between the text (or image) and the bridging context:

$$g_q = \tanh(\text{Linear}_1(Q_v, B))$$
$$g_k = \tanh(\text{Linear}_2(K_t, B)) \quad (10)$$

Following this, we acquire the image-aware text features $M^t \in \mathbb{R}^{n_t \times D}$ using the gated cross-attention:

$$M^t = \text{CrossAttention}(Q_v', K_t', V_t) \quad (11)$$

## 3.6 Classifier

In addition to the image-aware text features derived from Eqn. 11, we also incorporate each token's part-of-speech embeddings $M^{pos} \in \mathbb{R}^{n_t \times d_1}$, and positional embeddings $M^p \in \mathbb{R}^{n_t \times d_1}$, to enrich the information available for decoding. For MNER, a widely used CRF layer [1, 21] is employed to predict label sequences. For MRE and MJERE, we propose a word-pair contrastive layer to predict word-pair labels [45].

**CRF layer** Initially, we employ a multi-layer perceptron to integrate the three channels of features:

$$M = \text{MLP}_1(M^t; M^{pos}; M^p) \in \mathbb{R}^{n_t \times d_2} \tag{12}$$

Subsequently, we feed $M = \{M_1, \ldots, M_{n_t}\}$ into a standard CRF layer to derive the final distribution $P(y|t)$. The training loss for the input sequence $t$ with gold labels $y^*$ is measured using the negative log-likelihood (NLL):

$$\mathcal{L}_{main} = -\log P(y^*|t) \tag{13}$$

**Word-Pair Contrastive layer** For each word pair $(t_i, t_j)$, we initially employ three multi-layer perceptrons to separately obtain the three channels of pair-wise features:

$$X_{i,j}^l = \text{MLP}^l(M_i^l; M_j^l; M_i^l - M_j^l) \in \mathbb{R}^{d_2} \tag{14}$$

where $l \in \{t, pos, p\}$ represents the different feature channels. Following this, we use only the text features to generate the initial distribution $P_{i,j}^t$ and incorporate the three channels of features to derive the enhanced distribution $P_{i,j}^{t,pos,p}$ over the label set:

$$P_{i,j}^t = \text{Softmax}(\text{MLP}_2(X_{i,j}^t)) \tag{15}$$

$$P_{i,j}^{t,pos,p} = \text{Softmax}(\text{MLP}_3(X_{i,j}^t; X_{i,j}^{pos}; X_{i,j}^p)) \tag{16}$$

The final distribution is refined by contrasting the two distributions:

$$P_{i,j}^{final} = \text{Softmax}(P_{i,j}^{t,pos,p} + \lambda \log \frac{P_{i,j}^{t,pos,p}}{P_{i,j}^t}) \tag{17}$$

where $\lambda$ is the refinement scale. We use the cross-entropy loss for the input sequence $t$ and the gold word-pair labels $y^*$:

$$\mathcal{L}_{main} = -\sum_{i=1}^{n_t} \sum_{j=1}^{n_t} y_{i,j}^* \log(P_{ij}^{final}) \tag{18}$$

### 3.7 Training

In summary, our framework incorporates two self-supervised learning tasks and one supervised learning task, resulting in three loss functions. Considering the semantic loss $\mathcal{L}_{semantic}$, the modality loss $\mathcal{L}_{modality}$ from Sec. 3.4, and the prediction loss $\mathcal{L}_{main}$ from Sec. 3.6, the final loss function is defined as follows:

$$\mathcal{L} = \alpha \mathcal{L}_{semantic} + \beta \mathcal{L}_{modality} + \mathcal{L}_{main} \tag{19}$$

where $\alpha, \beta$ are hyperparameters.

## 4 EXPERIMENTS

We conduct our experiments on four MIE datasets and compare our method with a number of previous approaches.

### 4.1 Datasets

Experiments are conducted on 2 MNER datasets, 1 MRE dataset and 1 MJERE dataset: Twitter-15 [48] and Twitter-17 [44] for MNER[1], MNRE[2] [52] for MRE, and MJERE[3] [45] for MJERE, noting that the last dataset and the task share the same name. Statistical details of these datasets are shown in Table 1.

[1] The datasets are available at https://github.com/jefferyYu/UMT, https://github.com/Multimodal-NER/RpBERT

[2] https://github.com/thecharm/MNRE

[3] https://github.com/YuanLi95/EEGA-for-JMERE

**Table 1: Size of the datasets in numbers of tweets.**

| Dataset | Train | Val | Test |
|---------|-------|-----|------|
| Twitter-15 | 4,000 | 1,000 | 3,257 |
| Twitter-17 | 3,373 | 723 | 723 |
| MNRE | 12,247 | 1,624 | 1,614 |
| MJERE | 3,617 | 495 | 474 |

**Table 2: Prompts employed for LLaVA-1.5.**

Please generate a clear and concise textual description for this image sourced from a Twitter post. Describe this image in English as detailed as possible and avoid repetition in your explanation.

### 4.2 Implementation Details

**Model Configuration** In order to fairly compare with the previous approaches[13, 17, 40–43, 45, 46, 51], we use BERT-based model with the dimension of 768 as the textual encoder. For visual encoder, we experiment with the ViTB/32 in CLIP[4] and ResNet152 models as potential alternatives. We find ResNet152 with the dimension of 2048 to provide the most effective and consistent results in our settings. To generate the texual descriptive contexts of images, we use LLaVA-1.5 [26], a newly proposed visual instruction-tuned, large multimodal model[5]. The prompt we employ is shown in Table 2 and the sampling temperature is set to 1.0. We apply the spaCy[6] library of the en_core_web_sm version to parse the given sentence and obtain each word's part of speech. The dimensions of positional embeddings and part-of-speech embeddings are 100.

**Training Configuration** Shap-CA is trained by Pytorch on single NVIDIA RTX 3090 GPU. During training, the model is finetuned by AdamW [29] optimizer with a warmup cosine scheduler of ratio 0.05 and a batch size of 28. We use the grid search to find the learning rate over $[1 \times 10^{-5}, 5 \times 10^{-4}]$. The learning rate of encoders is set to $5 \times 10^{-5}$. The learning rates of other modules are set to $3 \times 10^{-4}$, $1 \times 10^{-4}$, $1 \times 10^{-4}$ and $1 \times 10^{-4}$ for Twitter-15, Twitter-17, MJERE and MNRE, respectively. The refinement scale $\lambda$ in Eqn. 17 is set to 1.0. An early stopping strategy is applied to determine the number of training epochs with a patience of 7. We choose the model performing the best on the validation set and evaluate it on the test set. We report the average results from 3 runs with different random seeds.

### 4.3 Baselines

We compare Shap-CA with previous state-of-the-art methods, which primarily fall into two categories. The first group of methods only consider text input, including BERT-CRF [13], MTB [7], ChatGPT and GPT-4. Secondly, we consider several latest multimodal methods, including OCSGA [42], UMGF [46], MEGA [51], MAF [43], ITA [40], CAT-MNER [41], MNER-QG [17], EEGA [45], and MQA [38].

[4] https://github.com/openai/CLIP

[5] https://github.com/haotian-liu/LLaVA

[6] https://github.com/explosion/spaCy

**Table 3: Performance comparison (F1 score) of our approach and state-of-the-art approaches on four MIE datasets. Following [45], on the MJERE dataset, we demonstrate both joint entity-relation extraction and named entity recognition results, denoted as MJERE$_j$, MJERE$_e$ respectively. T-15: Twitter-15, T-17: Twitter-17. † denotes the results are reproduced by us.**

| Modality | Method | MNER | | MJERE | | MRE |
|---|---|---|---|---|---|---|
| | | T-15 | T-17 | MJERE$_j$ | MJERE$_e$ | MNRE |
| Text | BERT-CRF | 71.81 | 83.44 | - | - | - |
| | MTB | - | - | - | - | 60.86 |
| | ChatGPT | 50.21 | 57.50 | 13.37† | 51.74† | 35.20 |
| | GPT4 | 57.98 | 66.61 | 21.51† | 56.63† | 42.11 |
| Text + Image | OCSGA | 72.92 | - | 49.64 | 76.07 | - |
| | UMGF | 74.85 | 85.51 | 51.45 | 76.75 | 65.29 |
| | MEGA | 72.35 | 84.39 | 53.18 | 77.22 | 66.41 |
| | MAF | 73.42 | 86.25 | 53.62 | 76.81 | - |
| | ITA | 75.60 | 85.72 | - | - | 66.89 |
| | CAT-MNER | 75.41 | 85.99 | - | - | - |
| | MNER-QG | 74.94 | 87.25 | - | - | - |
| | EEGA | - | - | 55.29 | 78.59 | - |
| | MQA | 50.6 | 62.6 | - | - | 61.6 |
| | Shap-CA w/o Context in Alignment | 74.87 | 86.19 | 54.07 | 77.74 | 66.29 |
| | Shap-CA w/o Context in Fusion | 75.24 | 87.31 | 53.35 | 77.49 | 66.41 |
| | Shap-CA w/o Shapley Value | 75.38 | 86.58 | 54.14 | 77.48 | 65.72 |
| | Shap-CA w/o word-pair Contrastive | - | - | 54.73 | 78.02 | 67.12 |
| | **Shap-CA** | **76.93** | **88.32** | **55.95** | **79.29** | **67.94** |

## 4.4 Main Results

Table 3 provides a comprehensive comparison of our proposed method with baseline approaches across various modalities, including text-based methods and previous state-of-the-art multimodal methods. Through this comparative analysis, we get several noteworthy findings as follows: (1) Superior Performance of Shap-CA: Our method, Shap-CA, demonstrates significant superiority across all datasets compared to baseline approaches. Notably, in tasks involving entity or relation extraction, such as Twitter-15, Twitter-17, MJERE$_e$, and MNRE, Shap-CA consistently outperforms the best competitor by substantial margins of 1.33%, 1.07%, 0.7%, and 1.05% in terms of F1 scores, respectively. These results underscore the effectiveness of our proposed Image-Context-Text paradigm in enhancing information extraction tasks. (2) Effective Handling of Complex Tasks: In the most challenging task, MJERE$_j$, which requires simultaneous extraction of entities and their associated relations, Shap-CA achieves the highest F1 score of 55.95%. This outcome highlights Shap-CA's efficacy in managing complex MIE scenarios, where accurate identification of both entities and relations is crucial. (3) Remarkable Adaptability: In contrast to other models that are typically specialized in one or two specific tasks, Shap-CA demonstrates exceptional adaptability by consistently delivering state-of-the-art performance across a diverse range of tasks, which underscores the model's ability to adapt to various environments and tasks, highlighting its flexibility in managing different information extraction challenges. (4) Enhanced Performance with Visual Information: Incorporating visual information from images generally enhances the performance of MIE when comparing state-of-the-art multimodal methods to their text-only counterparts. This observation suggests the integration of visual information provides a more holistic understanding of the input, underlining its value of improving the accuracy of information extraction tasks, particularly in scenarios where text contents alone may be insufficient for prediction. (5) Performance of Large Models: Contrary to the common belief that larger models possess superior generalization abilities, our experimental results show that methods based on LLMs (e.g., ChatGPT, GPT-4) and LMMs (e.g., MQA) perform less well than our proposed method and other previous multimodal approaches. This suggests that the current large models, despite their extensive capabilities, might not yet be fully optimized for information extraction tasks. This potential discrepancy could arise from their open-ended generative nature and pre-training scenarios that are misaligned with the specifics of MIE.

## 5 DISCUSSION AND ANALYSIS

### 5.1 Ablation Study

We conduct a comprehensive ablation study to further analyze the effectiveness of our method. The results of these model variants are presented in Table 3 and Table 4.

**Importance of Bridging Context** Our method fundamentally relies on the bridging context to enhance cross-modal comprehension and interpretation. To evaluate its effectiveness, we introduce two variants: (1) Shap-CA w/o Context in Alignment, which directly aligns image-text inputs based on the Shapley value, and (2) Shap-CA w/o Context in Fusion, which simply applies vanilla cross-attention fusion without the context involved. The results show that Shap-CA consistently outperforms these variations across all

**Table 4: Contribution of images to model performance.**

| Method | MJERE$_j$ | | | MJERE$_e$ | | |
|---|---|---|---|---|---|---|
| | P | R | F1 | P | R | F1 |
| Baseline | 50.89 | 53.93 | 52.36 | 74.09 | 78.75 | 76.35 |
| w/o Image | 51.81 | 56.29 | 53.96 | 74.31 | 80.30 | 77.19 |
| Shap-CA | **54.03** | **58.02** | **55.95** | **77.19** | **81.50** | **79.29** |

datasets. For instance, the F1 score decreases from 76.93% to 74.87% on the Twitter-15 dataset when the context in alignment is removed. This highlights that the textual context can effectively bridge the semantic and modality gaps between images and text, demonstrating its value in MIE.

**Role of Shapley Value** To understand the impact of the Shapley value in alignment, we replace it with InfoNCE [36], a widely used approach in the field of contrastive learning [5, 32, 35]. As shown in Table 3, Shap-CA apparently outperforms the variant without the Shapley Value on all datasets, which validates our proposed alignment method's ability to construct more coherent multimodal representations for MIE.

**Effectiveness of the Word-Pair Contrastive Layer** We further delve into the impact of our proposed word-pair contrastive layer by evaluating the performance of Shap-CA w/o word-pair Contrastive. The results show that contrasting the distributions before and after the integration of the three channels can effectively enhance the model's predictive power.

**Contribution of Images** Since the context of Shap-CA is generated from images, a natural question that arises is whether the context can completely replace the original image. In Table 2, we compare Shap-CA with its image-removed variant and the baseline on the MJERE dataset. The baseline approach simply concatenates the text and context before feeding it into a BERT-base model. Experimental results demonstrate that while the textual context information derived from the images is beneficial, it cannot fully replace the contribution of the images themselves to the model's performance.

## 5.2 Further Analysis

**How Alignments Affect Performance** In this section, we further discuss the influence of alignments on our model's performance on the MJERE dataset. Specifically, we examine two key coefficients, $\alpha$ and $\beta$ in Eqn. 19, which represent context-text alignment and context-image alignment, respectively. As shown in Figure 3, the model's performance exhibits a consistent pattern of variation in response to changes in $\alpha$ and $\beta$. The red line draws the performance trend of changing $\alpha$ when fixing $\beta$ as 0.4. $\alpha = 0$ means w/o context-text alignment, resulting in the poorest performance due to the inability to effectively bridge the semantic gap. When $\alpha = 0.2$, the model reaches its optimal performance. However, when $\alpha$ exceeds 0.2, the alignment task may interfere with the main task, leading to declining performance. The blue line draws the performance trend of $\beta$ with the fixed value of $\alpha$ (i.e., 0.2), which is similar to that of $\alpha$. The peak performance is reached when $\beta$ is 0.4.

**Impact of Different Context Generators** In practical situations, robustness to a range of contexts from different generators is crucial,

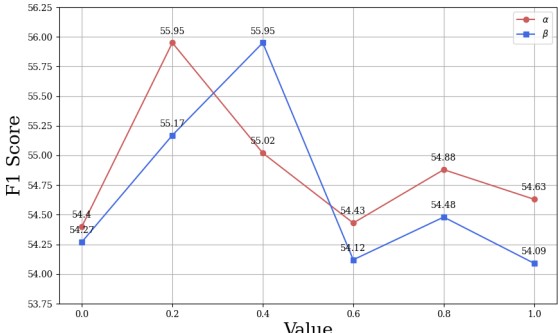

**Figure 3: Model performance on MJERE with different $\alpha$ or $\beta$**

**Table 5: Performance effect from the different context generators.**

| Generator | MJERE$_j$ | | | MJERE$_e$ | | |
|---|---|---|---|---|---|---|
| | P | R | F1 | P | R | F1 |
| BU | 54.11 | 56.92 | 55.48 | 76.57 | 80.92 | 78.69 |
| VinVL | 53.64 | 57.86 | 55.67 | 77.02 | 80.73 | 78.83 |
| BLIP-2 | **54.50** | 57.08 | 55.76 | 77.33 | 80.83 | 79.03 |
| LLaVA-1.5 | 54.03 | 58.02 | 55.95 | 77.19 | 81.50 | 79.29 |
| GPT-4 | 53.98 | **58.45** | **56.13** | **77.39** | **81.54** | **79.41** |

as these contexts may vary significantly in quality. To investigate this, we conduct experiments to assess the impact of employing various context generators. Specifically, we examine the results produced by five different context generators when applied to the MJERE dataset. These generators include traditional image captioning models: BU [3] and VinVL [47], and newly developed large multimodal models: BLIP-2 [24], LLaVA-1.5 [26], and GPT-4. As shown in Table 5, the performance disparity across these context generators is minimal. Notably, the use of GPT-4 as a context generator achieves the best overall performance. This observation implies that our method is robust to the variability inherent in different context generators, maintaining consistent performance regardless of the quality of the generated context.

**Model Scale** The scale of a model is a critical aspect in real-world applications. We present a comparative analysis of the total parameters and performance of our proposed method, Shap-CA, alongside other existing state-of-the-art multimodal approaches, whose results are achieved by their official implementations. As shown in Table 6, Shap-CA, with a parameter count of only 170.7M, is the most lightweight model among those evaluated, suggesting a reduced computational burden.

**Generalization** Table 7 presents a comparison of our method's generalization ability against several previous state-of-the-art approaches. This experiment is implemented by training on the source dataset while testing on the target dataset. For instance, T-17→T-15 denotes that the model is trained on the Twitter-17 dataset and then tested on the Twitter-15 dataset. As the results shown, our method

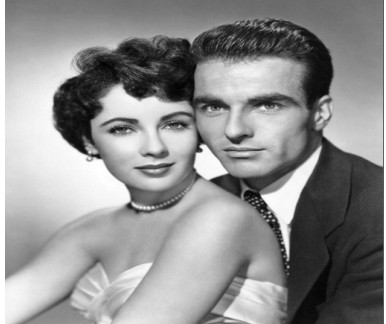

**Text:** Stephen Curry and Michael Jordan are both players who have/had 3 NBA Championships by age 30.

**Context:** The tweet is about a basketball game. There is a basketball player, holding up a ==trophy== and yelling in ==celebration==.

**Prediction:**
**Gold:** Stephen Curry$_{per}$, NBA Championships$_{misc}$, *awarded*
**EEGA:** Stephen Curry$_{per}$, NBA$_{misc}$, *present_in* ✗
**Ours:** Stephen Curry$_{per}$, NBA Championships$_{misc}$, *awarded* ✓

**Text:** Elizabeth Taylor and Montgomery Clift in A Place In the Sun, 1951 (dir. George Stevens).

**Context:** A black and white photo of a man and a woman. They are standing close to each other, with the man having a tie on. The man is looking at the camera while holding the woman, which gives an air of ==intimacy and connection== between the two.

**Prediction:**
**Gold:** Elizabeth Taylor$_{per}$, George Stevens$_{per}$, *couple*
**EEGA:** Elizabeth Taylor$_{per}$, George Stevens$_{per}$, *peer* ✗
**Ours:** Elizabeth Taylor$_{per}$, George Stevens$_{per}$, *couple* ✓

**Figure 4: Two cases of the predictions by EEGA and Shap-CA (ours).**

**Table 6: Model scale comparison. Results are achieved by their official implementations.**

| Method | Param. | T-15 | T-17 | MJERE$_j$ | MJERE$_e$ |
|---|---|---|---|---|---|
| UMT | 206.3M | 73.41 | 85.31 | - | - |
| CAT-MNER | 198.5M | 75.41 | 85.99 | - | - |
| EEGA | 179.7M | - | - | 55.29 | 78.59 |
| Shap-CA | 170.7M | **76.93** | **88.32** | **55.95** | **79.29** |

**Table 7: Comparison of the generalization ability. Results of other methods are from Li et al. [25], Wang et al. [41]**

| Method | T-17→T-15 | | | T-15→T-17 | | |
|---|---|---|---|---|---|---|
| | P | R | F1 | P | R | F1 |
| UMT | 64.67 | 63.59 | 64.13 | 67.80 | 55.23 | 60.87 |
| UMGF | 67.00 | 62.18 | 66.21 | 69.88 | 56.92 | 62.74 |
| FMIT | 66.72 | 69.73 | 68.19 | 70.65 | 59.22 | 64.43 |
| CAT-MNER | 74.86 | 63.01 | 68.43 | 70.69 | 59.44 | 64.58 |
| Shap-CA | 72.79 | 65.83 | **69.14** | 71.27 | 59.95 | **65.12** |

outperforms previous approaches by a very large margin in terms of F1 score, which highlights our method's strong generalization ability.

**Case Study** In Figure 4, we present two cases that highlight the effectiveness of our proposed method in bridging both semantic and modality gaps, challenging for EEGA [45] which follows a direct Image-Text interaction paradigm. The first case discusses Stephen Curry's achievements in the NBA. Although EEGA recognizes the

player, it incorrectly extracts "NBA" as an entity, leading to an inaccurate "present_in" relation between "Stephen Curry" and "NBA". In contrast, our method, utilizing the context featuring a "trophy" and "celebration", makes an accurate prediction, which denotes the context is helpful to mitigating the semantic and modality gaps. The second case is more nuanced, involving a scene with Elizabeth Taylor and George Stevens. Though EEGA can identify both persons, it fails to infer the intimate relationship implied by the image and text, predicting a mere "peer" relation. However, our method, leveraging the context that emphasizes "intimacy and connection", accurately predicts the relation "couple" between "Elizabeth Taylor" and "George Stevens".

## 6 CONCLUSION

In this paper, we introduce a novel paradigm of Image-Context-Text interaction to address the challenges posed by the semantic and modality gaps in conventional MIE approaches. In line with this paradigm, we propose a novel method, Shapley Value-Based Contrastive Alignment (Shap-CA), which performs both context-text and context-image alignments. Shap-CA first employs the Shapley value to access the individual contributions of contexts, texts, and images to the overall semantic or modality overlaps, and then applies a contrastive learning strategy to perform the alignment by maximizing the interactive contribution within context-text/image pairs and minimizing the influence across these pairs. Furthermore, Shap-CA incorporates an adaptive fusion module for selective cross-modal fusion. Our experiments demonstrate that Shap-CA significantly outperforms previous state-of-the-art approaches across four MIE datasets. In our analysis, we demonstrate Shap-CA's robust performance in the face of varying context quality, its high model efficiency and strong generalization ability.

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
