# OpenReview forum: "Shapley Value-based Contrastive Alignment for  Multimodal Information Extraction"
_acmmm.org/ACMMM/2024/Conference — MM2024 Poster_

### Official Review · Reviewer_caH7 · 2024-05-20

**Rating:** 4
**Confidence:** 2

**Summary:**

This paper proposes Image-Context-Text interaction with LLM for multimodal information extraction tasks to bridge semantic and modality gaps. A Shapley Value-based Contrastive Alignment method is proposed to align context-text and context-image pairs. Contrastive learning and adaptive fusion are used to enhance features. Experiments on four MIE datasets demonstrate its effectiveness.

**Strengths:**

1. The idea of generating context using LMM for bridging the semantic and modality gap is interesting.
2. The introduction of Shapley Value-based Contrastive Alignment is novel.
3. Experimental results achieve SOTA.
4. This paper is well-written and well-organized.

**Limitations:**

1. The context generated by LMMs can be seen as a kind of wild knowledge, which may be unfair to compare the proposed methods with previous methods directly. It would be better to integrate the context to text modality of previous methods for comparison.
2. Are the backbones of the compared methods the same in Table 1? It would be better to list the backbone.
3. Is the context needed in inference?

**Suitability:**

3

---

### Official Review · Reviewer_RNXc · 2024-05-23

**Rating:** 5
**Confidence:** 2

**Summary:**

This paper adopts the context as a bridge for the image-text interaction. Furthermore, a Shapley Value-based Contrastive Alignment is proposed to align the context-text and context-image pairs.

**Strengths:**

1. The idea of image-context-text interaction is novel and effective;
2. The paper is well-organized and written.

**Limitations:**

1. The performance improvements are not significant compared with previous works shown in Table 3, I'm not familiar with MIE, is that acceptable?
2. The figures in the paper are better to use vector images.

**Suitability:**

3

---

### Official Review · Reviewer_ETQ2 · 2024-05-25

**Rating:** 5
**Confidence:** 3

**Summary:**

This paper introduces a method for multimodal information extraction. It adopts LMM to generate a context for each image, which serves as a bridge between image and text to narrow both the semantic and modality gaps. Subsequently, it applies the Shapley value concept from cooperative game theory to refine the connections between modalities and adaptively fuse them. Experiments on four popular datasets validate the effectiveness and generalization of the proposed method.

**Strengths:**

1.	The paper is well-motivated. Using captions of images as contexts to bridge the semantic and modality gaps intuitively makes sense.
2.	It is clever to apply Shapley value concept from game theory. The experiments demonstrate its superiority compared to contrastive learning in aligning modalities.
3.	The comprehensive experimental results sufficiently demonstrate the advantages of the proposed method.

**Limitations:**

1.	The necessity of using LMMs as the context generator is not particularly convincing. It can be observed in Tab. 5 that the improvements brought by LMMs are limited. Comparing LMMs with more advanced expert models (such as OFA) might strengthen the results.
2.	Lack of visualization elements (such as heatmap) on how the generated contexts improve the model’s perceptions of images and texts.
3.	It appears that there is an oversight in line 728. The authors may have mistakenly written Table 4 as Table 2.

**Suitability:**

3

---

### Official Review · Reviewer_qjzp · 2024-05-29

**Rating:** 2
**Confidence:** 4

**Summary:**

The Shapley Value-based Contrastive Alignment (Shap-CA) method introduces a new paradigm of Image-Context-Text interaction for multimodal information extraction. Shap-CA utilizes large multimodal models to generate contexts and introduces the Shapley Value concept to evaluate the individual contribution of each context to mitigate semantic gaps.

**Strengths:**

This work is the first study to utilize Shapley Value for multimodal alignment via contrastive learning, referencing knowledge from game theory to multimodal tasks.

**Limitations:**

1.The motivation for introducing the Adaptive Fusion module is less clear, which is not strongly associated with the use of LLM-generated contexts as a bridge and the innovativeness of the Shapley Value. Further explanation is needed from the authors.
2.The text are concatenated with context and fed into BERT for textual feature encoding in Eqn. 1, which can cause potential interference between the feature encodings of the text and context due to the sequential contextual relevance of BERT itself. Does this affect the calculation of the cosine semantic similarity between the two in Eqn. 4.
3. The performance enhancement of this work is not particularly significant. Specifically, several competitive works, including ITA, MNER-QG, and EEGA, have not utilized a LMM. Table 3 presents a comparison of LMMs that is limited to MQA, and it lacks a performance benchmark for LLAVA 1.5 or other LMMs, which are utilized for context generation.
4. Please carefully check the details in the paper. Specifically, the symbols corresponding to part-of-speech embeddings and positional embeddings in 3.6 are written backwards. This will affect the reader's understanding.

**Suitability:**

3

---

### Meta-Review · Area_Chair_bgWZ · 2024-07-03

**Recommendation:** Accept (Poster)
**Confidence:** 3

**Metareview:**

The paper's primary strength lies in its novel application of Shapley Value for multimodal alignment via contrastive learning. This innovative approach, bridging concepts from game theory with multimodal tasks, has the potential to contribute significantly to the field of multimodal information extraction. Additionally, reviewer ETQ2 highlighted the intuitive nature of using image captions as contexts to bridge semantic and modality gaps.
However, several limitations were raised by reviewers. Most notably, reviewer qjzp, who provided the most detailed analysis, raised concerns about the limited performance improvements, especially considering the use of LMMs. This reviewer also pointed out potential technical issues, such as interference between text and context encodings, and questioned the motivation behind the Adaptive Fusion module. These technical concerns are particularly impactful.
After careful consideration of the strengths and weaknesses, I recommend Acceptance for this submission.